# Breast Cancer Drug Resistance: Overcoming the Challenge by Capitalizing on MicroRNA and Tumor Microenvironment Interplay

**DOI:** 10.3390/cancers13153691

**Published:** 2021-07-22

**Authors:** Giulia Cosentino, Ilaria Plantamura, Elda Tagliabue, Marilena V. Iorio, Alessandra Cataldo

**Affiliations:** Molecular Taregting Unit, Research Department, Fondazione IRCCS Istituto Nazionale dei Tumori di Milano, 20133 Milan, Italy; giulia.cosentino@istitutotumori.mi.it (G.C.); ilaria.plantamura@istitutotumori.mi.it (I.P.); elda.tagliabue@istitutotumori.mi.it (E.T.); marilena.iorio@istitutotumori.mi.it (M.V.I.)

**Keywords:** breast cancer, microRNAs, microenvironment, drug response

## Abstract

**Simple Summary:**

The cross-talk between neoplastic cells and microenvironment is known to play a crucial role in tumor development as well as in the phenomenon of resistance to anticancer therapies. MicroRNAs, involved in the pathogenesis of human tumors, are among the molecules exploited in this aberrant cross-talk. Indeed, microRNAs play a crucial function both in the induction of pro-tumoral traits in stromal cells and in the stroma-mediated fueling of tumor aggressiveness. Here, we reviewed the most recent literature concerning the role of miRNAs in shaping the tumor microenvironment, and the consequences on responsiveness to therapies.

**Abstract:**

The clinical management of breast cancer reaches new frontiers every day. However, the number of drug resistant cases is still high, and, currently, this constitutes one of the major challenges that cancer research has to face. For instance, 50% of women affected with HER2 positive breast cancer presents or acquires resistance to trastuzumab. Moreover, for patients affected with triple negative breast cancer, standard chemotherapy is still the fist-line therapy, and often patients become resistant to treatments. Tumor microenvironment plays a crucial role in this context. Indeed, cancer-associated stromal cells deliver oncogenic cues to the tumor and vice versa to escape exogenous insults. It is well known that microRNAs are among the molecules exploited in this aberrant crosstalk. Indeed, microRNAs play a crucial function both in the induction of pro-tumoral traits in stromal cells and in the stroma-mediated fueling of tumor aggressiveness. Here, we summarize the most recent literature regarding the involvement of miRNAs in the crosstalk between tumor and stromal cells and their capability to modulate tumor microenvironment characteristics. All up-to-date findings suggest that microRNAs in the TME could serve both to reverse malignant phenotype of stromal cells, modulating response to therapy, and as predictive/prognostic biomarkers.

## 1. Introduction

Overcoming tumor drug resistance is one of the major obstacles to obtaining a long-term cure. Patients often develop resistance to treatment, forcing them to change therapeutic strategy, which might in turn result in resistance phenomenon. In this review, we mainly focus our attention on breast cancer (BC) disease, which is the most common disease and the second-highest cancer-related cause of mortality in women. Breast cancer is a highly heterogeneous disease, subdivided into three main groups: hormone receptor positive (HR+), human epidermal growth factor receptor 2 positive (HER2+) and triple negative breast cancer (TNBC). In recent years, the increased awareness of women thanks to the incessant flow of information and the improvement in early detection has led to impressive results in terms of BC prevention and cure; however, recurrence, metastasis and therapy resistance still impact overall survival of BCs [1]. It is well known that there are two main types of resistance in cancer: (1) inherent or intrinsic resistance, meaning that insensitivity exists before treatment, and (2) acquired resistance, which appears following an initial positive response [2]. Consequently, it is crucial to study the mechanisms involved in drug-resistant signal transduction to overcome them.

### 1.1. MicroRNAs

MicroRNAs (miRNAs) are small non-coding RNAs that repress translation through interaction with 3′ untranslated regions (3′UTRs) of specific mRNA targets [3,4]. MiRNAs regulate different cellular functions such as development, differentiation and growth. Moreover, these small molecules are implicated in many human diseases, and their alterations are involved in the initiation and progression of human cancer. Current studies reveal that miRNAs also play a role in the process of drug resistance in cancer, even though the molecular mechanisms that contribute to sustaining this phenomenon are numerous and still unclear [5,6].

An important miRNA feature is the tissue-specific expression and/or mechanism of action, which can be exploited to better understand the evolution of the disease [7]. Moreover, knowing the miRNA tissue of origin could help in identifying a circulating miRNA as a biomarker for a specific disease.

However, there is an increasing effort to develop efficient strategies to deliver a miRNA-based therapy. In general, miRNA mimics are used to reintroduce oncosuppressive miRNAs in the target cells, while oncogenic miRNAs are inhibited by anti-miRs. Since these oligonucleotides are not stable in the extracellular compartment, they have to be either chemically modified or encapsulated into nanoparticles. Numerous are the compounds and formulations that have been designed over the last decade, and the most successful are currently being tested in clinical trials [8,9].

Nevertheless, side or off-target effects are still major issues when using miRNAs as therapeutic intervention.

An important example was described in 2016 by Yoo et al. The authors delivered miR-10b-inhibitory aminated magnetic nanoparticles with low-dose doxorubicin in in vivo models of metastatic breast cancer and successfully achieved complete and persistent regression with no evidence of systemic toxicity [10].

MiRNAs can benefit breast cancer clinical management serving also as easily detectable biomarkers to predict response to therapy. Indeed, analysis of tissue or circulating miRNA in patient’s serum or plasma can help in identifying patients likely to respond and change the therapeutic strategy for those already resistant [11].

To date, several miRNAs have been identified to be involved in breast cancer drug resistance; indeed, it was shown that some tumor suppressor miRNAs are downregulated in chemotherapy-resistant BC, and upregulation of these miRNAs leads to reversal of chemotherapy resistance. For example, reduction of miR-128 contributes to doxorubicin resistance in breast cancers [12]. In 2016, Gao M. et al. discovered in both in vitro and in vivo experiments that miR-145 sensitizes breast cancer to doxorubicin by suppressing the ABC transporter multidrug resistance-associated protein 1 (MRP1) and suggested the development of MRP1 inhibitors [13].

Furthermore, downregulation of oncogenic miR-181 and miR-663 resulted in improvement of the sensitivity of doxorubicin-resistant cells [14]. Moreover, in MDA-MB-231 cells, the phenomenon of chemoresistance was accompanied by the downregulation of heparan sulfate proteoglycan 2 (HSPG2), a matrix scaffold protein and target of miR-663 [15]. A miRNA signature that discriminated cisplatin-resistant BC cells versus sensitive BC cells showed that miR-146a, miR-221/222, miR-200b and miR-200c were significantly deregulated in resistant breast cancer cells [16]. In recent years, our group intensively studied the role of miR-302b in cisplatin sensitivity in triple negative BC cells [17,18]. Indeed, we demonstrated that miR-302b upregulation is able to increase cell sensitivity to cisplatin, acting on the oncogenic E2F Transcription Factor/YinYang1/Integrinα6 (E2F/YY1/ITGA6) axis. In addition, miR-129-3p was upregulated in docetaxel-resistant cells and contributed to docetaxel resistance in breast cancer cells; in contrast, other miRNAs, such as miR-100 and miR-30c were downregulated in paclitaxel-resistant BC cells [14,15,16].

Moreover, it is well known that HER2-positive BCs are mainly treated with monoclonal antibody trastuzumab, which significantly ameliorates the outcome of patients; however, 50 percent of them develop resistance. There is plenty of evidence that miRNAs also play a role in this scenario. MiR-21 and miR-221 have been reported to induce trastuzumab resistance and metastasis of HER2-positive BC [19]. MiR-200c, which is downregulated in trastuzumab-resistant cells, restores trastuzumab sensitivity and suppresses invasion of BC cells by targeting a mediator of transforming growth factor beta (TGF-β) signaling [20]. Moreover, we recently reported that miR-205 enhances trastuzumab sensitivity in HER2-positive BC through the downregulation of HER3 [21].

Overall, the role of miRNAs has been extensively studied in the intrinsic and acquired drug resistance in neoplastic cells, and it is currently well recognized how this phenomenon can be also mediated by components of the surrounding environment, where several type of cells are able to counteract drug response [22,23].

### 1.2. The Tumor Microenvironment

The microenvironment—or stroma—is a connective tissue consisting of a complex network of collagens, proteoglycans and glycoproteins—called the extracellular matrix (ECM)—and different type of cells. In the normal breast, the molecular composition and structure of the ECM changes in response to hormonal cues and significantly influences important cellular events, including proliferation, differentiation and gene expression [24]. The cellular component mostly includes fibroblasts, immune cells, endothelial cells and adipocytes. Each cell compartment provides a unique contribution in supporting the adjacent epithelium: Fibroblasts are involved in ECM synthesis and remodeling, local immune cells protect from infections and other insults, endothelial cells assemble blood vessels and adipocytes provide energy reserves [25,26,27].

Mutations in either epithelium or stroma can lead to an altered intercellular crosstalk and, eventually, to tumorigenesis [28]. Malignant cells are also able to induce the adjacent stromal cells to acquire a tumor-supporting behavior by secreting soluble factors such as epidermal (EGF), platelet-derived (PDGF), vascular endothelial (VEGF) and fibroblast (FGFs) growth factors and TGF-β or by releasing microvesicles, such as exosomes, to vehicle RNA molecules or proteins. For example, cancer-associated fibroblasts (CAFs) present increased proliferation, motility and secretion of ECM proteins, growth factors, cytokines and proteases; immune cells tend to have an immunosuppressive phenotype; endothelial cells are induced to actively provide oxygen and nutrient supply and for a means of tumor dissemination; and, finally, cancer-associated adipocytes (CAAs), similarly to CAFs, release pro-proliferative and pro-invasive stimuli and reprogram cell metabolism [29,30,31]. Heterogeneity in breast cancer can also be found in the tumor microenvironment, affecting both tumor progression and response to therapy [32]. For example, the presence of spatially and functionally distinct subsets of CAFs has been described in the breast TME, which also influence the composition of the immune compartment, thus impacting patient prognosis [33,34]. Accordingly, both quality and quantity of immune cells present at or recruited to the tumor site differ according to the molecular subtype of BC and predict response to therapy [35]. Moreover, therapy itself has an immunomodulatory activity, either damping or accentuating anti-tumor activity [36,37,38].

As mentioned above, the communication between tumor and stroma cells could be mediated by RNA molecules like miRNAs. Indeed, miRNAs encapsulated in microvesicles, exosomes or lipoproteins can be delivered by tumor cells to other cells in the tumor microenvironment, and vice versa, or in the blood, thus modulating the phenotypic switch to cancer-associated cells, locally or at distant sites [39,40,41].

The dynamic role of miRNAs in BC and its surrounding environment suggests the use of miRNAs as a therapeutic strategy, helping to prevent development of therapy resistance. Indeed, considering the impact of the tumor microenvironment on the biology of breast malignances, it is extremely relevant to design new and more efficient therapeutic strategies.

Here we dissect the literature regarding the role of miRNAs in TME cells and their exchange with cancer cells through exosome delivery.

## 2. Role of miRNAs in Modulating Breast Cancer Microenvironment

### 2.1. Fibroblasts

Fibroblasts, the major cellular component of the stroma, physiologically support and protect the epithelia by synthesizing and remodeling the ECM when insults occur. In tumors, fibroblasts can acquire aberrant properties and sustain progression and dissemination, dialoguing with cancer and other stromal cells; these fibroblasts are designated as CAFs and differ from their normal counterparts (NFs) both functionally and genetically [42,43]. For example, miRNA expression is deregulated in CAFs vs. NFs, contributing in shaping an oncogenic phenotype and acting as a messenger in the crosstalk with the neighboring cells [44,45]. Accordingly, the CAF secretome, including miRNA-containing exosomes, and ECM composition can also affect tumor response to treatment. Indeed, our group recently demonstrated that CAF-like cells, induced by the tumor-mediated upregulation of a specific metastamiR, miR-9, are also able to affect TNBC responsiveness to cisplatin [46,47].

In ER + BCs, the specific subset of CD63 + CAFs induces resistance to the endocrine therapy tamoxifen via exosomal miR-22 [48]. In 2017, Sansone P. et al. demonstrated that microvesicles containing the CAF-released miR-221 contribute to the formation of hormone therapy-resistant stromal-tumor niches, promoting the generation of CD133^hi^ cancer stem cells [49]. Another work by Liu Y. et al. shows that miR-29a is involved in tumor-CAFs interaction by interfering with p38 signal transducer and activator of transcription 1 (p38-STAT1) signaling, thus affecting breast cancer cell growth, drug resistance and metastasis [50]. In 2020, the same group focused on another CAF-released miRNA, miR-3613-3p, demonstrating its role as oncogene, promoting breast cancer drug resistance, ROS production and metastasis through the inhibition of suppressor of cytokine signaling 2 (SOCS2) [51].

Furthermore, it was demonstrated that miRNA-105 secreted by breast cancer is able to activate MYC signaling in CAFs to promote a metabolic program. The authors reported that miR-105 influences the metabolism in the environment, altering it according to nutrient levels. Indeed, when nutrients are sufficient, miR-105-reprogrammed CAFs mediate glucose and glutamine metabolism to stimulate adjacent cancer cells. When nutrient levels are low these CAFs convert lactic acid and ammonium into energy-rich metabolites. In conclusion, the miR-105-mediated metabolic reprogramming induces tumor growth by acting on stromal cells [52].

### 2.2. Immune Cells

#### 2.2.1. Macrophages

Breast cancer-associated macrophages (TAMs) are the major immune component in the BC microenvironment. TAMs are predominately derived from circulating monocytes at the tumor site, and they are able to promote tumor growth, angiogenesis, metastases and drug resistance by producing cytokines, enzymes and factors [53]. Generally, macrophages have been classified as M1, with an anti-tumor activity inducing Th1 response, or M2, activating Th2 response and promoting tumor growth and invasion [54]. M1 macrophages are stimulated by lipopolysaccharides (LPSs), interferon gamma (IFN-γ), and tumor necrosis factor (TNF-α) and trigger the secretion of pro-inflammatory cytokines including interleukin-1 (IL-1), IL-6, IL-12, IL-23 and TNF-α, reactive nitrogen and oxygen intermediates (RNI, ROI) [55]. TAMs usually have an M2 phenotype, are responsive to anti-inflammatory factors such as IL-4, IL-13 and IL-10 and secrete C-C motif chemokine ligands 2, 17 and 22 (CCL2, CCL17 and CCL22) and transforming-factor β (TGF-β), thus inducing tumor growth and dissemination. Moreover, M2 macrophages have been associated with resistance to chemotherapeutic treatments [56].

It has largely demonstrated that macrophage polarization is affected by miRNAs, thus influencing immune response in the tumor microenvironment [57,58]. The first demonstration of miRNA involvement in macrophage polarization was reported in 2012 by Cay et al., who confirmed that M2 macrophages can be reprogrammed in the pro-inflammatory M1 phenotype by miR-155 [59]. Briefly, the authors compared miRNA expression profiles in bone marrow-derived macrophages (BMDMs), polarized in M1 by LPS/IFN-ɣ or in M2 by IL-4. The results showed that miR-155 was the most upregulated when BMDMs were switched into the M1 phenotype, giving to miR-155 an important role in promoting macrophage polarization to the M1 phenotype. Moreover, it was reported that miR-125b induces activation of macrophages and consequently immune response by targeting interferon regulatory factor 4 (IRF-4). Indeed, macrophages over-expressing miR-125b better stimulated T-cell activation [60].

Many studies in breast cancer models have demonstrated the role of miRNAs in the phenotypic switch of macrophages in the TME. For example, miR-19a-3p overexpression inhibited the M2 phenotype, inducing a downregulation of fos-related antigen 1 (FRA-1)/STAT3 signaling and a reduction of migration and invasion of breast cancer cells. The authors also showed that miR-19a-3p impairs the metastatization capability by regulating TAM in an in vivo breast cancer model [61].

It is also known that miRNAs can be transferred to TME cells from tumor cells, influencing their behavior, and vice versa. Jang J.Y. and colleagues demonstrated that miR-16 was upregulated in BC cells by epigallocatechin gallate (EGCG), a polyphenol present in green tea with an anti-tumoral effect, and it was transferred to TAM via exosomes. This led to an inhibition of M2 polarization and an anti-tumor effect [62].

Exosomal miR-503 induced M2 polarization in microglia of brain metastases, upregulating immunosuppressive factors and reducing T-cell proliferation. This phenomenon was caused by the loss of lncRNA XIST, which induces miR-503 secretion thus promoting brain metastasis in breast cancer. The authors identified fludarabine, a synthetic lethal drug able to eradicate XIST^low^ tumor cells in the brain [63].

Concerning the uptake from tumor cells, miR-375, highly expressed in breast cancer, is released in an LDL-bound entity during apoptosis and captured by macrophages via CD36. In macrophages, miR-375 promotes migration and infiltration into the tumor in both in vitro and in vivo models of breast cancer [64].

Another study showed that miR-100 expression maintains the TAM phenotype by targeting mechanistic target of rapamycin (mTOR) signaling. Expression of miR-100 in TAMs induced secretion of immunosuppressive cytokines such as IL1Ra. Importantly, the combined intratumoral injection of 4T1 breast cancer cells and miR-100 antagomiR reduced tumor growth and metastases and promoted response to cisplatin [65].

Moreover, in TNBC patients, low expression of miR-149 correlated with macrophage infiltration and reduction of overall survival. Following the reintroduction of miR-149 in TNBC in vitro models, the recruitment of human monocytic THP-1 cells and primary human macrophages was inhibited. Accordingly, in in vivo models, miR-149 expression reduced lung metastasis formation, impairing the M2 macrophage recruitment at the tumor site [66].

Finally, miR-146 and miR-222 have been found to regulate M2 polarization and the recruitment of TAMs [67].

All these findings support the idea that microRNAs, modulating the behavior of TAM, could contribute to reducing therapy resistance.

#### 2.2.2. Myeloid-Derived Suppressor Cells

MDSCs are typically immunosuppressive cells and are classified into monocytic MDSCs (CD11b + CD14 + HLA-DR^−^^/low^ CD15^−^), expressing inducible nitric oxide synthase (iNOS) and producing nitric oxide (NO), and granulocytic MDSCs (CD11b + CD14 − HLA-DR^low^^/^^−^ CD15+), generating reactive oxygen species (ROS) and arginase-1 [68]. Moreover, MDSCs secrete immunosuppressive cytokines such as IL-10 and TGF-b, regulating T-cells and natural killer (NK) cell function. MDSC activation also enhances PD-L1 expression and immune suppression.

MDSCs have an important role in development and progression of breast cancer; for this reason, they are a potential therapeutic target [69]. It was found that miRNAs regulate myeloid cell metabolism and switch to the immunosuppressive phenotype, thus modulating their influence on tumor cell behavior.

MiR-155 loss in deficient breast tumors leads to an increase of MDSC infiltration in the tumor microenvironment. The data show that CCAAT/enhancer-binding protein beta (C/EBP-β), a known miR-155 target, induces the release of several cytokines to recruit MDSCs. The authors demonstrated that miR-155 inhibition plays a dual role in breast cancer and tumor microenvironment, leading to a decrease of in vivo tumor growth, which is restored by the loss of miR-155 in the microenvironment. They suggest that miR-155 antitumor therapy could have an unfavorable effect due to the recruitment of immunosuppressive cells [70].

In another study, it was found that the novel regulatory prostaglandin E2 (PGE2)/miR-10a/5′ AMP-activated protein kinase (AMPK) axis is involved in chemotherapy-induced immune resistance. Doxorubicin-resistant tumors induce prostaglandin E2 secretion, promoting MDSC expansion and polarization and a consequent upregulation of miR-10. This miRNA can trigger AMPK signaling to promote expansion and activation of MDSCs. Thus, PGE2/miR-10a/AMPK axis might be targeted for treatment of chemotherapy-resistant tumors [71].

Treatment with doxorubicin in 4T1 breast-tumor-bearing mice induces activation of IL-13R + miR-126a + MDSC, promoting lung metastasis through miR-126a release in exosomes. Using a specific miR inhibitor, a significant reduction of miR-126a expression in MDSCs and in MDSC-derived exosomes was found. Moreover, miR-126a transfection in Th2-polarized T-cells enhanced the secretion of IL-4 and IL-13. Thus, miR-126a transfer from MDSCs to T-cells may promote the induction of Th2 cells. These findings indicate that miR-126a is induced by doxorubicin treatment contributing to MDSC-exosome-mediated induction of Th2 T-cells, prompting angiogenesis and metastasis [72].

Moreover, other miRNAs were found to be involved in MDSC activation and accumulation in breast tumor tissue. For example, miR-494 is upregulated in MDSCs by tumor-derived factors and regulates cell activation by targeting PTEN and the triggering of Akt signaling. By silencing miR-494, MDSC activity is significantly reversed, and 4T1 murine breast cancer tumor growth and metastasis are inhibited [73]. Recently, Jang M. et al. demonstrated that miR-9 and miR-181 enhance the amplification of immature early-stage MDSCs (eMDSCs) with a consequent strong immunosuppression in T-cells both in mice and humans. This phenomenon is due to the activation of JAK/STAT signaling by two different crucial factors, suppressor of cytokine signaling 3 (SOCS3) and protein inhibitor of activated STAT3 (PIAS3), targeted by miR-9 and miR-181, respectively. Moreover, high levels of these two miRNAs were detected in tumor-derived exosomes, and miRNA-mediated activation of eMDSCs is attenuated by blocking the exosome release [74]. All these findings support the idea that miRNA reintroduction or suppression could be a potential therapeutic strategy to revert the immunosuppressive environment in breast cancer.

#### 2.2.3. T-Cells

In most cases, the breast tumor microenvironment is infiltrated by tumor-infiltrating lymphocytes (TILs) [75]. It has been reported that presence of different lymphocyte populations such as CD8+ cytotoxic, forkhead box P3-positive (FOXP3+) regulatory and CD4+ helper and follicular T-cells is associated with patient prognosis and response to therapies. CD8+ cytotoxic T-lymphocytes can eradicate cancer cells, leading to higher response to therapies and better prognosis, but often these cells become exhausted, exhibiting reduced proliferation and decreased production of IFN-γ, and are unable to kill tumor cells [76]. In the context of the tumor microenvironment, miRNAs can be employed to efficiently modulate T-cell differentiation and function. MiR-149-3p was predicted to target programmed cell death protein 1 (PD-1), T-cell immunoglobulin and mucin domain 3 (TIM-3), B- and T-lymphocyte-associated (BTLA) and Foxp1 T-cell inhibitory receptors. MiR-149-3p mimic transfection mediates increased T-cell activation with an increase of IL-2, TNF-α and IFN-γ secretion. Moreover, the miR-149-3p mimic reintroduction enhances the capacity of CD8+ T-cells to kill 4T1 mouse breast tumor cells. Thus, the authors conclude that miR-149-3p can be a potential antitumor immunotherapeutic agent in breast cancer, reverting CD8+ T-cell exhaustion [77]. Moreover, it is known that miR-155 dramatically impacts both innate and adaptive immune processes, including inflammation, antigen presentation, T-cell differentiation, cytokine production and T regulatory cell (Treg) functions. Indeed, the suppressor of cytokine signaling 1 (SOCS1) is a critical regulator of immune cell function and an evolutionarily conserved target of miR-155 in breast cancer cells [78]. Another study reveals that miR-155 promotes CD8+ T-cell antitumor function. The authors demonstrated that CD8+ T-cell differentiation is regulated through the miR-155–Phf19 (PHD Finger Protein 19)–PRC2 (polycomb repressive complex 2) axis, paving the way for new methods for potentiating cancer immunotherapy through epigenetic reprogramming of CD8+ T-cell fate [79]. Hong B.S. and colleagues have recently demonstrated that miR-204-5p can be a key regulator of the breast tumor immune microenvironment by modulating different types of myeloid and T-cells. They found that miR-204-5p-overexpressing in vivo tumors showed a substantial shift in the composition of various immune cells in the tumor microenvironment. Particularly, there were significant reductions in MDSCs, macrophages and natural killer (NK) cells in the miR-204-5p-overexpressing tumors. On the other hand, CD4+ T-cells and CD8+ T-cells, including regulatory T-cells, showed significantly increased prevalence in the microenvironment of the miR-204-5p-overexpressing tumors [80]. A few years ago, it was reported that miR-21 was highly expressed in Tregs in tumor tissues from a murine breast cancer model. Silencing miR-21 significantly reduces the proliferation of Tregs in vitro and, as shown by the adoptive cell-transfer assay, alters the enrichment of Tregs in the tumor, thus improving the antitumor effect of CD8(+) T-cells in a murine BC model. This effect is due to the fact that miR-21 silencing enhanced the expression of its target PTEN and subsequently altered the activation of the Akt pathway, which is responsible for reduced proliferation activity of Tregs. Moreover, it was found that miR-21 expression is higher in Tregs from BC patient tissues. Thus, miR-21 is likely able to modulate T-cell immunity by regulating a distinct subset of Tregs. This might represent a new immunotherapeutic strategy [81]. Similarly, silencing of miR-126 could significantly reduce the induction of Tregs in vitro. Indeed, it was observed that miR-126 silencing downregulated the expression of Foxp3 in Tregs and decreased expression of cytotoxic T-lymphocyte antigen 4 (CTLA-4) and glucocorticoid-induced TNFR-related protein (GITR), as well as IL-10 and TGF-β, counteracting its immunosuppressive ability. Silencing of miR-126 also altered the activation of the PI3K/Akt pathway. Finally, as above, miR-126 silencing impaired the suppressive function of Tregs in vivo and promoted an antitumor effect of CD8(+) T-cells in an adoptive cell transfer assay using a murine breast cancer model [82].

#### 2.2.4. Dendritic Cells

Dendritic cells (DCs) are antigen-presenting cells that are able to induce an anti-tumor T-cell response. Indeed, DCs represent a connection between innate and adaptive immunity and could be an effective strategy for immunotherapy [83]. Usually, tumors can prevent antigen presentation and inhibit the activation of tumor-specific immunity. On the other hand, tumor-derived factors can alter DC maturation helping tumor growth and generating “pro-tumor” inflammation. In recent years, many studies have focused on the possibility of introducing DC-based immunotherapy and DC vaccines as potent immunotherapies to treat cancer, inducing and enhancing T-cell-mediated immune responses [84].

As described above, miRNAs can be involved in the regulation of the immune tumor microenvironment and response to therapy, including DC activation. Some studies have been performed to evaluate miRNAs as a booster for DC immunotherapies.

MiR-155 is considered a good miRNA candidate. Boosting the expression of miR-155 may significantly improve the efficacy of DC-based immunotherapies for breast cancer. It was found that miR-155 is a master regulator of DC function in breast cancer, including maturation, cytokine secretion, migration toward lymph nodes, and activation of T-cells [85]. More recently, the same group generated a DC vaccine using miR-155 overexpressing DCs. The results showed that vaccine enhances antitumor immunity against established breast cancers in mice, increasing effector T-cells, suppressing tumor growth and drastically reducing lung metastasis formation [86].

Another study revealed that let-7b delivery could reactivate tumor-infiltrating DCs by acting as a TLR-7 agonist and suppressing IL-10 production in vitro. In a breast cancer mouse model, let-7b delivery efficiently reprogrammed the functions of both tumor-infiltrating DCs and tumor-associated macrophages, counteracting the suppressive tumor microenvironment and inhibiting tumor growth [87].

Finally, it was reported that miRNA-5119 was downregulated in splenic DCs from mouse breast-cancer-bearing mice. Zhang M. et al. suggested a novel therapeutic approach by using miRNA-5119 mimic-engineered DC vaccines to regulate inhibitory receptors and enhance anti-tumor immune response in a mouse model of breast cancer. Thus, DCs engineered to express a miR-5119 mimic downregulated programmed death-ligand 1 (PD-L1) and treatment of 4T1 breast-tumor-bearing mice with miR-5119 mimic-engineered DC vaccine reduced T-cell exhaustion and suppressed mouse breast tumor homograft growth [88].

### 2.3. Endothelial Cells

The endothelium is one of the most important components of the microenvironment. Endothelial cells (ECs) participate in forming a single layer that regulates exchanges between the bloodstream and the adjacent tissues. In pathological conditions, such as cancerous diseases, these cells participate in tumor cell growth, proliferation, dissemination and metastasis [89].

As expected, miRNAs can also target specific angiogenic genes, modulating tube formation and metastasis, in tumor cells. For example, Liang Z. et al. showed that miR-206 inhibits tumor invasion and angiogenesis and also decreases VEGF, MAPK3 and SRY-box transcription factor 9 (SOX9) levels in TNBC cells lines [90]. This paper demonstrated for the first time the involvement of miRNA-206 in TNBC invasion and angiogenesis and suggested miR-206 as an efficient tool for TNBC therapy. In the same period, our group proposed miR-9 and miR-200c as therapeutic tools to inhibit the process of vascular lacunae formation in this TNBC subtype [91]. Particularly, in MDA-MB-231 and MDA-MB-157 TNBC cell lines, miR-9 and miR-200 promoted and inhibited, respectively, the formation of vascular-like structures both in in vitro and in vivo experiments. These results demonstrated that miR-9 and miR-200 play opposite roles in the regulation of the vasculogenic ability of TNBC, promoting and suppressing platelet-derived growth factor receptor beta (PDGFRβ), respectively. Moreover, our data support the possibility to therapeutically exploit miR-9 and miR-200 to inhibit the process of vascular lacunae formation in TNBC. Another miRNA, miR-29b, is implicated in the phenomenon of angiogenesis and it is downregulated in BC versus normal tissues [92,93].

Recently, it was observed that miR-20a, a member of the miR-17-92 cluster, is a possible angiogenic marker in patients with invasive breast carcinoma [94]. Gines Luengo-Gil’s data demonstrated that a TNBC MDA-MB-231 cell line transfected with miR-20a mimic showed endothelial tube formation capability. Moreover, these angiogenic effects were abrogated by treatment with an anti-VEGF drug, aflibercept. Thus, expression of miR-20a in BC promoted an angiogenic pattern that consists of the presence of large vessels, anomalous glomeruloid microvascular proliferations and high VEGFA expression. It is known that VEGFA has a crucial role in BC tumorigenesis through its effects on tumor angiogenesis [95]. A recent paper has demonstrated miR-17-92’s role in endothelial proliferation in the context of VEGF-dependent tumor angiogenesis. These experimental data point to the potential value of miR-17-92 as a target for blocking VEGFA-stimulated tumor angiogenesis [96]. It was also found that miR-140-5p counteracts tumor invasion and angiogenesis ability of breast cancer cells both in vitro and in vivo by targeting VEGF-A [97]. Moreover, miR-205 targets VEGF-A and FGF2, promoting chemosensitivity in breast cancer cells [98]. Finally, more recently, Song Y. and colleagues demonstrated that miR-152-3p modulates paclitaxel resistance in BC by targeting endothelial PAS domain-containing protein 1 (EPAS1) [99]. In particular, they showed that miR-152-3p is downmodulated in BC compared to normal tissues, and its reintroduction, or EPAS1 silencing, enhanced MCF-7/TAX cell sensitivity to paclitaxel. Finally, overexpression of EPAS1 reverts the paclitaxel sensitivity induced by miR-152-3p. In 2012, it was revealed that endogenous miR-126 modulates endothelial cell recruitment to metastatic breast cancer cells. In particular, it suppresses metastatic endothelial recruitment, metastatic angiogenesis and metastatic colonization by targeting pro-angiogenic genes such as insulin like growth factor binding protein 2 (IGFBP2), cytoplasmic phosphatidylinositol transfer protein 1 (PITPNC1) and MER proto-oncogene, tyrosine kinase (MERTK) [100]. Several studies focused on the role of miRNAs in the angiogenic process, evaluating the crosstalk between endothelial and tumor cells. In 2014, it was demonstrated that miR-105 has an important role in metastasis formation; indeed, exosome-mediated transfer of breast cancer-secreted miR-105 in an endothelial monolayer promotes the destruction of the tight junction by targeting zonula occludens-1 (ZO-1) [101]. Furthermore, miR-105 reintroduction in non-metastatic cells induces metastasis formation, and in breast cancer patients increased circulating miR-105 levels can be detected at the pre-metastatic stage and correlate with metastasis occurrence. Moreover, Di Modica M. et al. have demonstrated that BC cells released miR-939 in exosomes; once internalized in endothelial cells, the miRNA’s direct inhibition of VE-cadherin enhanced endothelial monolayer permeability, eventually favoring the trans-endothelial migration of tumor cells [22]. Accordingly, miR-939 was found to be associated with worse prognosis in TNBCs. Similarly, another study showed that miR-153 directly targeted angiopoietin 1 (ANG1) in BC cells and inhibited the migration and the tube formation of endothelial cells [102]. A functional axis that induces the crosstalk between tumor and endothelial cells was identified by McCann J.V. et al. This axis involved TGF-β, miR-30c and Serpine1, which suppressed vascularization and breast cancer growth [103]. MiR-7, in addition to its tumor-intrinsic suppressive role, blocks the homing and migration of endothelial cells under BC cell conditioning [104]. Indeed, miR-7 transfected in endothelial cells significantly inhibited their proliferative, chemotactic and angiogenic-like homing characteristics, especially in response to chemoattractant factors produced by aggressive breast cancer cells. Recently, a novel endothelial-tumor cell interaction, mediated by the endothelial miR-125a/let-7e-IL-6 signaling axis, was found [105]. In particular, cisplatin treatment induced a downregulation of miR-125a and let-7e expression and an upregulation of IL-6 levels in endothelial cells. Conditioned medium from cisplatin-treated endothelial cells induces a higher formation of vasculogenic mimicry in BC cells, eventually causing drug resistance and metastasis. Furthermore, miR-199b-5p was downregulated in breast cancer cells and in endothelial cells [106]. Lin X. et al. identified that ALK1 is directly targeted by miR-199b-5p, which was able to inhibit tumor growth and angiogenesis both in vitro and in vivo. Moreover, reintroduction of miR-199b-5p in endothelial cells inhibits the formation of capillary-like tubular structures and migration. Endothelial cells can also release miRNAs in the exosome as cues to modulate tumor cell behavior. For example, in BC tumor cells, miR-503, produced and released by endothelial cells, impairs tumor growth by targeting cyclin D2 and D3 (CCND2 and CCND3) [107]. Additionally, BC patients receiving neoadjuvant chemotherapy showed increased circulating levels of miR-503. To prove the origin of this miRNA, endothelial cells were treated with the same chemotherapeutic agents used in the clinic, and a drastic increase of miR-503 release was observed in the exosome.

### 2.4. Adipocytes

Adipocytes play an active role in the tumor microenvironment [108]. In the breast cancer microenvironment, adipocytes represent the primary cellular components, due to the fact that, compared to other organs, this tissue is characterized by a high concentration of fatty cells [109]. Cancer-associated adipocytes (CAAs) have an important role in enhancing breast cancer progression; indeed, emerging evidence reveals that tumor progression is also promoted through the constant communication between tumor cells and adipocytes.

As mentioned, miRNAs can be delivered from tumor cells to the TME and vice versa, and the delivery of onco-miRNAs such as miRNA-144, miRNA-126 and miRNA-155 from BC cells to adipocytes in the BC microenvironment via exosomes results in the conversion of resident adipocytes into CAAs [110,111].

In particular, tumor–adipocyte interaction leads to the release in exosomes of miR-144 and miR-126, which induce metabolic reprogramming, thus sustaining tumor progression.

Additionally, exosomal miRNA-155 is able to promote lipolysis in adipocytes and facilitate an aggressive phenotype of tumor cells. MiR-155 directly targets peroxisome proliferator activated receptor gamma (PPARγ), which is involved in lipid accumulation. These results show that cancer-cell-secreted miR-155 promotes differentiation and alters the metabolism of surrounding adipocytes by downregulating PPARγ expression, accelerating a cancer-lipolytic process that has been described to be a phenomenon associated with tumor progression.

In another study, it was demonstrated that mmu-miR-5112 is upregulated in 3T3-L1 adipocytes indirectly co-cultured with breast cancer cells. In these cells there was also the upregulation of inflammation-related genes including Il-6 and pentraxin 3 (PTX3). Mmu-miR-5112 targets Cpeb1, which is a negative regulator of Il-6. Thus, mmu-miR-5112 reintroduced in adipocytes led to an upregulation of *Il6*, confirming the role of the miRNA in promoting the acquisition of CAA inflammatory phenotypes [112].

Genome-wide analysis revealed 98 miRNAs differentially expressed in BC cells after co-culture with mature adipocytes compared to BC cells alone. Results revealed that miR-3184-5p and miR-181c-3p were the most upregulated and downregulated, respectively. MiR-3184-5p and miR-181c-3p targets forkhead box P4 (FOXP4), a gene involved in cancer growth and metastasis, and peroxisome proliferator-activated receptor alpha (PPARγ), known to be an oncogene. Thus, miR-3184-5p inhibitor and miR-181c-3p mimic co-transfected in adipocytes impaired cell proliferation, invasion and epithelial–mesenchymal transition (EMT) in BC cells, suggesting a strong involvement of miRNA expression in adipocytes in the regulation of neoplastic transformation of BC cells [113].

Another group showed that the interaction of human adipocyte stem cells (hASCs), pre-adipocytes and mature adipocytes with BC cells leads to increased release of pro-inflammatory cytokines such as IL6, IL8, IFN-IP10, C-C motif chemokine ligand 2 (CCL2) and CCL5 by BC cells. The interaction of cancer cells and adipocytes or pro-inflammatory cytokines promoted mammosphere-forming cells and cells expressing stem-like markers. The authors also investigated the mechanism behind this phenomenon, and they found that the interaction of BC cells and immature adipocytes induced miR-302b upregulation, mediated by Src, through the enhancement of SRY-box transcription factor 2 (SOX2), sustaining the production of pro-inflammatory cytokines and thus BC stem cell renewal [114].

Lastly, it was demonstrated that miR-140, secreted in exosomes from pre-adipocytes, affected the neighboring BC cells. Results demonstrated that 3T3L1 cells present an increase of miR-140 and a consequently downregulation of its target SOX9 following antitumor treatment. Moreover, exosomes derived from 3T3L1 cells have detectable expression levels of SOX9 protein; this phenomenon is abrogated upon treatment, inducing miR-140 expression. Thus, treated pre-adipocytes are able to release exosomes carrying a high level of miR-140, which can affect nearby tumor cells by targeting the SOX9 pathway [115]. Taken together, these results highlight the importance of studying the impact of adipocyte miRNAs on breast neoplastic transformation, tumor progression and therapy response.

## 3. Conclusions

In recent years, emerging evidence concerning the role of miRNAs in the crosstalk between tumor and stroma cells supports the idea that a miRNA-based intervention might not only affect tumor cell behaviors but also the characteristics of the whole tumor environment, comprising stroma cells that support tumor growth and aggressiveness. Moreover, all findings reported in the literature support the idea that miRNAs could also influence the mechanisms of drug resistance by mediating the TME remodeling (Figure 1). This information paves the way for a miRNA-based therapy combined with standard treatment in breast and other cancers to solve one of the significant concerns for the scientific community. Table 1 summarizes miRNA localization and roles described in this review.

## Figures and Tables

**Figure 1 cancers-13-03691-f001:**
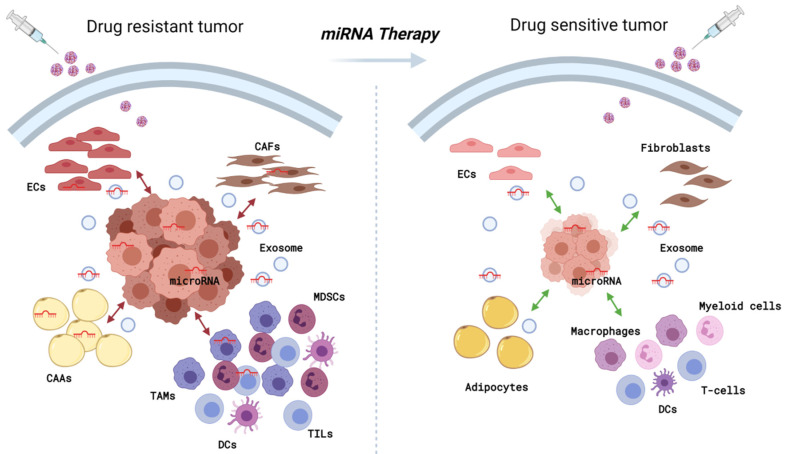
Hypothetical miRNA-based therapy model with a microRNA that, by affecting cancer cells and their environment, has anti-tumoral effects. The relevance of miRNA activity in shaping the breast TME (**left** panel), supports the idea of miRNA modulation as a therapeutic strategy. Thus, disrupting aberrant crosstalk and reverting malignant phenotypes would help to overcome or prevent therapy resistance (**right** panel). The figure was created with BioRender.com (accessed on 11 June 2021).

**Table 1 cancers-13-03691-t001:** Cell-specific functions of microRNAs.

MicroRNA Expression	Cell Type	Roles
hsa-miR-9	Fibroblasts (CAFs), MDSCs, tumor cells	Tumor growth, drug resistance [46,47]. MDSC activation and immunosuppression [69]. Promotion of vascular lacunae formation [82]
hsa-miR-22	Fibroblast (CAFs)	Drug resistance [39]
hsa-miR-221	Fibroblast (CAFs)	Drug resistance [40]
hsa-miR-29a	Fibroblast (CAFs)	Cell growth, drug resistance and metastasis [41]
hsa-miR-3613-3p	Fibroblast (CAFs)	Drug resistance, ROS production and metastasis [42]
hsa-miRNA-105	Fibroblast (CAFs)	Metabolic reprogramming [43]
hsa-miR-155	Macrophages, MDSCs, T-cells, Dendritic cells.	Inflammatory role [50,61,69,70,76,77,103]
hsa-miR-125b	Macrophages	Immune response [51]
hsa-miR-19a-3p	Macrophages	M2 phenotype inhibition [52]
hsa-miR-16	Macrophages	M2 phenotype inhibition [53]
hsa-miR-503	Macrophages	M2 phenotype induction [54]
hsa-miR-375	Macrophages	Migration and infiltration into the tumor [55]
hsa-miR-100	Macrophages	TAM phenotype Maintenance and immunosuppression [56]
hsa-miR-149-5p	Macrophages	M2 phenotype inhibition and lung metastasis reduction [57]
hsa-miR-146	Macrophages	M2 phenotype induction [58]
hsa-miR-222	Macrophages	Inhibition of TAM recruitment and of tumor growth [58]
hsa-miR-10	Tumor cells	Activation of MDSC and drug resistance [62]
hsa-miR-126a	MDSCs, T-cells	Immunosuppression and induction of Th2 cells [63]
hsa-miR-494	MDSCs	MDSC activation [64]
hsa-miR-181	MDSCs	MDSC activation and immunosuppression [69]
hsa-miR-149-3p	T cells	Reversion of CD8+ T-cell exhaustion [68]
hsa-miR-204-5p	Tumor cells	Reduction of MDSCs, macrophages, and natural killer (NK) cells and increase of CD8+ and CD4+ cells [71]
hsa-miR-21	T cells	Treg proliferation [72]
hsa-miR-126	T cells	Treg proliferation [73]
hsa-let-7b	Dendritic cells and macrophages	Reprogramming of the functions of both tumor-infiltrating DCs and tumor-associated macrophages [78]
hsa-miR-5119	Dendritic cells	Reduction of T-cell exhaustion and suppressed breast tumor growth [79]
hsa-miR-206	Tumor cells	Inhibition of tumor invasion and angiogenesis [81]
hsa-miR-200c	Tumor cells	Inhibition of vascular lacunae formation [82]
hsa-miR-29b	Endothelial cells	Anti-angiogenesis [83,84]
hsa-miR-20a	Tumor cells	Angiogenesis [85,86]
hsa-miR-17-92	Tumor cells	Angiogenesis [87]
hsa-miR-205	Tumor cells	Anti-angiogenesis and drug sensitivity [89]
hsa-miR-152-3p	Tumor cells	Anti-angiogenesis and drug sensitivity [90]
hsa-miR-126	Tumor cells, adipocytes	Inhibition of endothelial cell recruitment [91], metabolic reprogramming of adipocytes [102]
hsa-miR-140	Tumor cells	Anti-angiogenesis and tumor inhibition [88]
hsa-miR-105	Endothelial cells	Angiogenesis and metastasis formation [92]
hsa-miR-939	Tumor cells, endothelial cells	Endothelial monolayer permeability and trans-endothelial migration of tumor cells [93]
hsa-miR-153	Tumor cells	Inhibition of migration and of tube formation of endothelial cells [94]
hsa-miR-30c	Tumor cells, endothelial cells	Suppression of vascularization and inhibition of tumor growth [95]
hsa-miR-7	Endothelial cells	Inhibition of proliferative, chemotactic and angiogenic-like homing characteristics of endothelial cells [96]
hsa-miR-125a	Endothelial cells	Downregulated by cisplatin treatment, induction of vasculogenic mimicry [97]
hsa-let-7e	Endothelial cells	Downregulated by cisplatin treatment, induction of vasculogenic mimicry [97]
hsa-miR-199b-5p	Tumor cells, endothelial cells	Inhibition of tumor growth, angiogenesis, formation of capillary-like tubular structures and migration [98]
hsa-miR-503	Endothelial cells	Inhibition of tumor growth [99]
hsa-miR-155	Tumor cells, adipocytes	Differentiation and alteration of the metabolism of surrounding adipocytes, tumor progression [103]
hsa-miR-144	Tumor cells, adipocytes	Metabolic reprogramming of adipocytes [102]
mmu-miR-5112	Adipocytes	Acquisition of cancer-associated-adipocytes inflammatory phenotypes [104]
hsa-miR-3184-5p	Adipocytes	Regulation of neoplastic transformation of BC cells [105]
hsa-miR-181c-3p	Adipocytes	Regulation of neoplastic transformation of BC cells [105]
hsa-miR-302b	Adipocytes	Production of pro-inflammatory cytokines [106]
hsa-miR-140	Adipocytes	Regulation of neoplastic transformation of BC cells [107]

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
