# Peer review of "Breast Cancer Drug Resistance: Overcoming the Challenge by Capitalizing on MicroRNA and Tumor Microenvironment Interplay"

_cancers, 2021, doi:10.3390/cancers13153691_

Round 1

Reviewer 1 Report

This is an interesting review that provides an overview of recent microRNA findings in breast cancer that suggest a cell-type specific function in treatment resistance. The authors explore cancer cell-intrinsic and TME-driven activity of these microRNAs.

Major concerns

The abstract suggest a more direct relation between microRNA function and treatment resistance to trastuzumab in HER2+ cases and chemotherapy in TNBC. While the first section of the review touches more directly on this, the rest of the review does not.

The same microRNA is discussed in different context with different roles in cancer cells, fibroblasts and/or immune cells (e.g., miR-21 and miR-155). Some of the discussed microRNAs such as miR-9, miR-126, miR-145, miR-200 seems to have very different and predominantly expression patterns in neurons, endothelial cells, smooth muscle cells and epithelial cells, respectively. A more explicit discussion on cell type-specific expression of these miRNAs in relation to breast cancer subtypes would be helpful. Also if specific subset of tumor specifically expressed on of this microRNA in a cell type (e.g. cancer cells) whereas in others is in a different cell type (e.g. macrophages) or whether the same microRNA plays different and overlapping roles in different cell types of the same tumor. A table or figure summarizing the cell type-specific roles of miRNA and whether there is or not overlap with other cell types in the same tumor would be helpful.

The authors indicate that microRNA-based therapies could help overcome treatment resistance. A bit more of a discussion on what technologies and approaches may be used to increase or decrease miRNA activity and current challenges to implement these in the clinic will be helpful.

Minor concerns

It is not clear what Fig.1 tries to convey in terms of microRNA-based therapy. It is not clear what microRNA(s) are targets in what cell type(s). it seems that the effect of this therapy is “normalization” of the TME.

The authors seem to provide a reasonable numbers of recent reference, but the review is not comprehensive. For example, there are report of interaction between miR-10b and doxorubicin in different in vivo breast cancer models, including https://www.ncbi.nlm.nih.gov/pmc/articles/PMC4609288/ that are not discussed.

Please consider another work for upmodulation perhaps upregulation is more appropriate in this context.

Author Response

The abstract suggest a more direct relation between microRNA function and treatment resistance to trastuzumab in HER2+ cases and chemotherapy in TNBC. While the first section of the review touches more directly on this, the rest of the review does not.

Thanks to the reviewer for highlighting this concern. To date, there is not much evidence of a direct effect of microRNAs in the tumor microenvironment modulating trastuzumab and chemotherapy response. However it is known that microRNA expression can induce resistance or responsiveness to these drugs, and that stromal cells activation can also influence responsiveness to chemotherapy and targeted-therapy, impacting patients’ prognosis. Here, we speculate that modulating microRNA expression in tumor microenvironment cells, thus reverting their pro-tumoral phenotype, it is possible toovercome resistance to therapy. However, we added two references concerning the fundamental role of tumor microenvironment in response to trastuzumab and chemotherapy (line 112). 

A more explicit discussion on cell type-specific expression of these miRNAs in relation to breast cancer subtypes would be helpful.

Thank you for your comment. We added sentences in the introduction to underline this important issue concerning the role of microRNAs in different cells and tissues. We specified and discussed that microRNAs are generally cell and tissue specific, and this aspect is important overall in the contest of drug response and therapy (lines 63-67).

Also if specific subset of tumor specifically expressed on of this microRNA in a cell type (e.g. cancer cells) whereas in others is in a different cell type (e.g. macrophages) or whether the same microRNA plays different and overlapping roles in different cell types of the same tumor. A table or figure summarizing the cell type-specific roles of miRNA and whether there is or not overlap with other cell types in the same tumor would be helpful.

Thank you to the reviewer for the comment and suggestion. In this review we mainly focused our attention on microRNAs in TME cells; surely their role in tumor cells could be the same or opposite. Off target and side effects on different cell types, as described in revised version (lines 77-78), are major issues when facing miRNA-based therapy in clinical management. However, inhibition of miR-9 could be a good strategy for miRNA-therapy because miR-9 is an oncogene both in tumor cells and in TME cell, as reported in the literature and in this review. Viceversa, miR-155 has been described as an oncogenic miRNA in breast cancer, while it has an opposite role in TME cells. Indeed miR-155 is reported to be an inflammatory miRNA upregulated, for example, in anti-tumoral macrophage M1 (Cai X. et al 2012). Here, we also reported the dual role of miR-155 in the different cell types; accordingly, in this case, it is reasonable to find a cell-specific targeting strategy to deliver this microRNA.

We added a table (Table 1) that summarizes all microRNAs described in this review and their known localization and roles (line 569).  

The authors indicate that microRNA-based therapies could help overcome treatment resistance. A bit more of a discussion on what technologies and approaches may be used to increase or decrease miRNA activity and current challenges to implement these in the clinic will be helpful.

Thanks to the reviewer for this suggestion. We added in the introduction some information concerning miRNA-based therapy (lines 71-78).

It is not clear what Fig.1 tries to convey in terms of microRNA-based therapy. It is not clear what microRNA(s) are targets in what cell type(s). it seems that the effect of this therapy is “normalization” of the TME.

Thank you to the reviewer for the comment. Figure 1 is a hypothetical miRNA-based therapy model with a microRNA that, affecting cancer cells and their environment, has anti-tumoral effects (e.g. anti-miR-9 option). We apologize if the figure is not clear, but in our opinion introducing the name of microRNAs in the different cell types might become confusing. We hope that our explication and sentence added in the figure legend render the figureclearer and suitable for this context.

The authors seem to provide a reasonable numbers of recent reference, but the review is not comprehensive. For example, there are report of interaction between miR-10b and doxorubicin in different in vivo breast cancer models, including https://www.ncbi.nlm.nih.gov/pmc/articles/PMC4609288/ that are not discussed.

Thank you to the reviewer for pointing out this reference. We added it in the text at lines 79-82.

Please consider another work for upmodulation perhaps upregulation is more appropriate in this context.

Thank you for the suggestion, we replaced “upmodulation” with “upregulation”.

Reviewer 2 Report

Cosentino et al. reviewed the regulatory roles of miRNAs in drug resistance/sensitivity and tumor microenvironment (TME) in breast cancers. Though a comprehensive list of research articles were compiled by the authors to demonstrate that miRNAs play pivotal roles in each aspect and the key results were discussed, little connection between the two was provided. The authors should provide a few miRNA examples that increase/decrease the drug resistance in breast cancer cells directly by regulating the TME or target genes known to be controlling the TME. In addition, the development of miRNA drugs is still in its infancy, the authors should discuss how miRNAs can be the targeted to sensitize the drug-resistant cancer cells. The authors should also discuss how miRNAs can serve as the biomarkers for breast cancer patients who may develop drug resistance to improve the clinical decisions and outcomes.

Author Response

Thank you to the reviewer for the comments and suggestions. Regarding the little connection between microRNAs in TME and the direct effect on tumor resistance, unfortunately, there is not much evidence of the effects of microRNAs in the tumor microenvironment modulating response to therapies in breast cancer. Here, we speculate that modulating microRNAs expression in tumor microenvironment cells, and reverting their pro-tumoral phenotype, could avoid resistance to therapy. Concerning the use of microRNAs in therapy and as biomarkers we added some information and discussion in the introduction (lines 68-81).

Round 2

Reviewer 1 Report

The authors have sufficiently addressed my concerns and suggestions.

Minor concern

The authors introduced two new instances of down- or up-modulation after agreeing to use down- or upregulation instead. See line 169-70, Line 218

Author Response

The authors introduced two new instances of down- or up-modulation after agreeing to use down- or upregulation instead. See line 169-70, Line 218

Dear reviewer, we apologize for the mistake. In this new version we corrected the sentences. Thank you for the kind suggestion.

Reviewer 2 Report

The authors have resolved all the issues and made the necessary changes to the manuscript. I am satisfied with their responses.

Author Response

Dear reviewer, we are glad that you find the final version of our manuscript suitable for publication, and thank you again for your precious suggestions.